# Feasibility of Bio–Coagulation Dewatering Followed by Bio–Oxidation Process for Treating Swine Wastewater

**DOI:** 10.3390/ijerph20042990

**Published:** 2023-02-08

**Authors:** Dejin Zhang, Weicheng Han, Yujun Zhou, Cheng Yan, Dianzhan Wang, Jianru Liang, Lixiang Zhou

**Affiliations:** 1Department of Environmental Engineering, College of Resources and Environmental Sciences, Nanjing Agricultural University, Nanjing 210095, China; 2Jiangsu Key Laboratory of Chemical Pollution Control and Resources Reuse, School of Environmental and Biological Engineering, Nanjing University of Science and Technology, Nanjing 210094, China

**Keywords:** bio–coagulation, dewatering, biodegradation, swine wastewater, pilot–scale experiment

## Abstract

The unsatisfactory performance of the conventional swine wastewater treatment is drawing increasing attention due to the large amount of refractory chemical oxygen demand (COD), nitrogen, and phosphorus attached to the suspended solids (SS). In this study, for the first time, a novel process based on bio–coagulation dewatering followed by a bio–oxidation (BDBO) system was developed to treat swine wastewater containing high–strength SS, COD, TN, and TP. Firstly, after the bio–coagulation process, the removal efficiencies of SS, COD, NH_3_–N, and TP reached as high as 99.94%, 98.09%, 61.19%, and 99.92%, respectively. Secondly, the filtrate of the bio–coagulation dewatering process was introduced into the subsequent bio–oxidation process, in which the residual COD and NH_3_–N were further biodegraded in a sequence batch reactor. In addition, the dewatering performance of the concentrated swine slurry was substantially improved, with the specific resistance to filtration decreasing from 17.0 × 10^12^ to 0.3 × 10^12^ m/kg. Moreover, the concentrated swine slurry was pressed and filtered into a semi–dry cake after pilot–scale bio–coagulation dewatering treatment. Finally, the concentrations of COD and NH_3_–N in the effluent after the BDBO process, ranging between 150–170 mg/L and 75–90 mg/L, met the relevant discharge standard. Compared to traditional treatments, the BDBO system has excellent large–scale potential for improving the treatment efficiency, shortening the operation period, and reducing the processing costs, and is emerging as a cost–effective alternative for the treatment of wastewater containing high concentrations of SS, COD, TN, and TP.

## 1. Introduction

Rapid urban expansion and economic development have contributed to a boom in pig farming. China is the largest consumer of pork worldwide and breeds more than 500 million pigs per year [1], making sustainable swine manure management a challenging issue, particularly in an era with carbon neutrality as a key developing objective [2]. Due to the high concentrations of suspended solids (SS), chemical oxygen demand (COD), nitrogen, and phosphorus, as well as their temporal variability [3], concentrated swine wastewater has become a severe environmental problem [4].

Generally, swine wastewater generated from scattering raising households is stored or stabilized in anaerobic lagoons before being applied as biofertilizer on cropland. In intensive pig industries, the associated large amounts of swine wastewater require a series of process equipment with the characteristics of large space and long retention times [5]. Referring to our previous study on the failure of swine wastewater treatment [6], the conventional combined anaerobic/aerobic process after solid–liquid separation might be unstable due to high concentrations of “refractory” COD, nitrogen, and phosphorus attached to suspended solids (SS). Only a rather small part of the total COD is biodegradable in swine wastewater [7], leading to the poor methanogenic performance of the anaerobic digestion process. Additionally, too low a biodegradable COD/N ratio, ranging between 4 and 6.5, does not meet the basic needs for complete denitrification [3]. Various combined processes based on fermentation, partial denitrification, partial nitrification, and anammox have been conceived and investigated to realize efficient biological nitrogen removal [8,9], undoubtedly resulting in an extension of the treatment process. Concerning phosphorus, approximately 4–10% is dissolved and 3–20% is linked to the biomass, while 60–85% is precipitated [10]. Methods for the recovery and reuse of phosphorus from swine wastewater mainly focus on chemical precipitation [11]; however, external chemicals need to be added to balance the ratios of the elements and to increase the pH of the system. To summarize, the above have inspired a novel idea of whether high concentrations of “refractory” COD, nitrogen, and phosphorus attached to SS could be substantially removed to relieve the stress of the subsequent biological processes, simultaneously achieving the goals of improving the treatment efficiency, shortening the operation period, and reducing the processing cost.

Coagulation is one of the most commonly applied techniques to achieve efficient solid–liquid separation in wastewater treatment [12]. In the coagulation process, small colloids suspended in water are destabilized after diminishing their surface charges by adding coagulants [13]; then, the aggregated solid particles can be removed by sedimentation, followed by filtration or flotation. Typically, the coagulation efficiency depends mainly on the selected coagulants. Chemical coagulants such as polyacrylamide (PAM), polymeric aluminum chloride (PAC), and polymeric ferric sulfate (PFS) have been applied to the solid–liquid separation of swine wastewater treatment [14,15]. Accordingly, the generated chemical sludge primarily contains sand and coagulants [16], which hinders its reclamation and poses potential environmental risks. Compared to traditional chemical coagulants, microbial bioflocculants, natural organic macromolecular substances produced by microorganisms, have attracted significant attention due to their advantages of biodegradability, non–toxicity, high efficiency, and cost–effectiveness [17]. Biological coagulation using *Acidithiobacillus* species is considered effective in removing pollutants (e.g., suspended solids, organic pollutants, and heavy metal ions) from wastewater at the laboratory scale [18]. During this process, a microbial mixture composed of *Acidithiobacillus ferrooxidans* (*A. ferrooxidans*), *Acidithiobacillus thiooxidans* (*A. thiooxidans*), and *Acidiphilium* sp., etc., could create favorable conditions for flocculation and the dewatering of sludge flocs. It is well known that *A. thiooxidans* is sensitive to various kinds of organic compounds, such as simple sugars, amino acids, and organic acids [19]. Swine wastewater is rich in organic compounds, which might contain substances that are toxic to chemoautotrophic bacteria and, thus, decrease the treatment efficiency. Moreover, a range of acid–tolerant heterotrophic microorganisms that are able to metabolize organic compounds as a source of energy and carbon for growth have been reported to build mutualism with autotrophic bacteria [20]. Therefore, a combined process with the coinoculation of autotrophic bacteria and heterotrophic microorganisms would be able to improve sludge dewaterability. In addition, the liquid phase produced from this process contains low concentrations of nitrogen and phosphorous, which can be removed by secondary biological treatments, such as a sequence batch reactor (SBR). Therefore, a new concept is proposed for treating swine wastewater with high–strength SS, COD, nitrogen, and phosphorus through the bio–coagulation dewatering followed by bio–oxidation (BDBO) system. Moreover, in contrast to chemical conditioning, the bio–coagulation dewatering process could result in more organic matter, TN, and TP being retained in the resulting sludge cake. Therefore, considering the chemical properties of sludge filtrate and dewatered sludge cakes, this novel BDBO system not only achieves highly efficient treatment efficiency and meets the discharge standards, but also significantly reduces the processing costs of the subsequent reutilization or disposal of dewatered sludge. In our previous studies, a microbial mixture with excellent flocculation ability was applied in bioleaching processes for improving municipal sludge dewaterability [21] and pre–treating swine wastewater (from Huizhou, Guangdong, China), followed by dewatering with a diaphragm press filter [22]. Although the effective removal of SS, COD, and TP after excrement dewatering was achieved, the removal percentage of NH_3_–N was only 32.3%. To the best of our knowledge, the feasibility and performance of the BDBO system in treating real swine wastewater have rarely been explored.

Herein, the feasibility of the BDBO system in treating real swine wastewater is demonstrated. The dewatering performance of swine wastewater after the biological coagulation process was comprehensively investigated, with the chemical coagulants being compared. Furthermore, the long–term operation of SBR was also explored to evaluate the subsequent biological oxidation efficiency. The results of this work are expected to provide a scientific basis for applying the BDBO system to the treatment of wastewater with high–strength SS, COD, nitrogen, and phosphorus.

## 2. Materials and Methods

### 2.1. Source of Swine Wastewater

The raw swine wastewater used in this study was sampled from a liquid–solid separator of a swine manure treatment plant in a pig farm located in Hunan City, Changsha Province, China. This large–scale pig farm has adopted the flushing system to remove swine manure and produces 400 tons of swine wastewater per day. The primary physicochemical characteristics of the obtained swine wastewater are summarized as follows: pH 7.20–8.14, SS 3040–4900 mg/L, COD 6440–11,290 mg/L, NH_3_–N 652.3–1044 mg/L, TN 721.3–1187 mg/L, and TP 55.5–148.1 mg/L.

### 2.2. Preparation of Inoculum

The Acidophilic heterotrophic bacterium *Acidiphilium* sp. JZ6 (CGMCC No. 11036) was cultivated in a WAYE medium [23], with 2.0 g/L of glucose serving as the carbon source. The Acidophilic chemoautotrophic bacterium *A. thiooxidans* TS6 (CGMCC No. 0759) was cultivated in a mineral salt (MS) medium [24], with 10 g/L of powdered sulfur (S^0^) serving as the energy source. The pH of the WAYE and MS media were adjusted to 2.0 and 3.0 by sulfuric acid, respectively. *Acidiphilium* sp. JZ6 and *A. thiooxidans* TS6 were inoculated into the pre–sterilized WAYE and MS media and then incubated at 180 rpm and 28 °C for 3–4 days until the bacterial cell density reached approximately 10^8^ cells/mL. In order to enrich strains of *Acidiphilium* sp. JZ6 and *A. thiooxidans* TS6, the aforementioned cultivation procedures were repeated twice. The inoculum for the bio–coagulation process was the mixture of *Acidiphilium* sp. JZ6 and *A. thiooxidans* TS6 at the volume ratio of 1:1.

### 2.3. Startup and Operation of BDBO System

#### 2.3.1. Bio–Coagulation

Bio–coagulation was evaluated using a series of 500 mL transparent PET bottles as batch reactors. First, 360 mL of raw swine wastewater was added to the bottles and settled down for 1 h to simulate the grit tanks in the swine manure treatment plant of the pig farm. Then, 40 mL of the above inoculum was added to the 500 mL transparent PET bottles at an inoculation size of 10% (*v*/*v*) [21]. Comparative experiments were carried out using conventional chemical coagulants, including PAM, PAC, and PFS. Based on the preliminary experiment, the dosage of PAM, PAC, and PFS was 0.0085%, 0.1%, and 0.1% of the total mass, resulting in final concentrations of 80, 1000, and 1000 mg/L, respectively. The control treatment was operated without adding the above inoculum or any chemical coagulators. In addition, 40 mL of water was added to the control and comparative experiments to keep all treatments at the same volume of 400 mL. The sedimentation rate of the conditioned swine wastewater was determined after being thoroughly mixed. The supernatant after sedimentation was withdrawn for the analysis of the pH, COD, NH_3_–N, TN, and TP, while the concentrated swine slurry at the bottom of the reactors was collected for the subsequent pilot–scale dewatering experiment.

#### 2.3.2. Dewatering

The dewatering performance of the swine wastewater after the bio–coagulation process was evaluated using 100 mL beakers as batch reactors. First, 50 mL of the above concentrated swine slurry and 5 mL of the above inoculum were added to the reactors, followed by either (i) 0.2% bioflocculant (patented product, owned by Nanjing BACT Environmental Solutions Co., Ltd., Nanjing, China), or (ii) 0.4% lime milk, or (iii) 0.2% bioflocculant and 0.4% lime milk, or (iv) 0.3% bioflocculant and 0.4% lime milk, or (v) 0.2% bioflocculant and 0.3% lime milk, or (vi) 0.2% bioflocculant and 0.5% lime milk. After being thoroughly stirred for 5 min, the specific resistance to filtration (SRF) of the swine slurry samples was determined.

Furthermore, a pilot–scale dewatering experiment was also carried out in a 15 L cylinder reactor (25 cm in diameter and 30 cm in height), as shown in Appendix A. The optimal parameters selected in this study were in accordance with the above results of the vacuum filter experiment. First, 1.3 L of the inoculum was added into 13 L of concentrated swine slurry at a stirring rate of 200 rpm for 10 min, followed by the addition of 25 g of bioflocculant and 50 g of lime milk. After being thoroughly stirred for 20 min, 14 L of the reactants was extracted by a pneumatic diaphragm pump and fed into a slag remover, which was reformed from a diaphragm pressure filter (Model: XMG10/800–UK). The pilot–scale dewatering experiment was conducted with a filtration area of 0.5 m^2^, a work pressure of 0.4–0.6 MPa, a working time of 30 min, and a pressure holding of 15 min. The extraction time, per liter, of the reactant was recorded to calculate the filter pressing rate during filtering.

#### 2.3.3. Bio–Oxidation

A 125 L cylindrical plastic bucket (40 cm in diameter and 100 cm in height) with an effective working volume of 90 L and a volumetric exchange ratio of 20–25% was used as the SBR to treat the filtrate collected from the dewatering process. Aerobic granular sludge (0.5–0.8 mm in diameter) collected from the CASS tank of the manure treatment in the pig farm was used as the seed sludge. The SBR was operated in a cycle period of 24 h: 0.5 h of feeding without stirring, 12 h of aerobic reaction, 3 h of sludge settling, 0.5 h of effluent withdrawal, and 8 h of idling. Aeration and mixing were carried out by air bubble diffusers placed at the bottom of the reactor with an oxygen supply of 0.08 m^3^/h. The temperature was maintained at 30 ± 5 °C using a water bath circulator with a heating system. All operation processes were automatically controlled by programmable timers. The mixed liquid suspended solids (MLSS) and sludge volume index (SVI) were maintained within the range of 4000–5000 mg/L and 68 to 80 mL/g, respectively, during a period of continuous operation.

### 2.4. Analytical Methods

Analyses of COD, SS, NH_3_–N, TN, TP, SVI, and MLSS were performed in accordance with APHA [25]. The sedimentation rate of the conditioned swine wastewater samples was determined by measuring the supernatant volume after 24 h of natural sedimentation and calculating the percentage of the supernatant volume to the total volume [26]. The SRF was determined by the Buchner funnel method under a vacuum pressure of 640 mm Hg [27]. The moisture content was determined by drying the sample to a constant weight at 105 °C. The organic matter content was measured according to the method described by Nelson and Sommers [28]. All of the tests and analyses were conducted in triplicate and the data are reported as the arithmetic mean ± standard deviation.

## 3. Results and Discussion

### 3.1. Bio–Coagulation Performance

As the sedimentation rate can be indirectly used to evaluate the dewaterability of the sewage sludge and the solid–liquid separation ability of the sludge system [21,29,30], the bio–coagulation performance of the swine wastewater can be characterized by the sedimentation rate to reflect its sedimentation capability. Figure 1 depicts a comparison of the coagulation performance of the swine wastewater by using the mixed microbial mixture with the three conventional chemical flocculants. The sedimentation rate of the raw swine wastewater was 18.87 ± 2.3%, indicating poor settleability. However, the sedimentation rates in those treatments with the addition of PAM, PAC, and PFS were significantly improved, to 71.70 ± 5.2%, 67.25 ± 1.1%, and 63.63 ± 3.7%, respectively. Most importantly, the sedimentation rate in the bio-coagulation group was further increased to 83.05 ± 3.1%. It is clearly demonstrated that the bio–coagulation garnered much better results by improving the sedimentation ability of the swine wastewater than chemical conditioning, thus showing a strong solid–liquid separation effect. The same phenomenon was noted by Liu et al. [21], who found that the bio–coagulation process induced by *A. thiooxidans* and *A. ferrooxidans* was relatively more effective than other chemical conditioning treatments in reducing the SRF of sewage sludge by 93.1%. Furthermore, it is worth noting that both a clear solid–liquid interface and supernatant were observed after the bio–coagulation process. Thus, the bio–coagulation process induced by bioflocculants of *Acidiphilium* sp. JZ6 and *A. thiooxidans* TS6 proved to be an efficient method to improve the sedimentation capability of swine wastewater.

Table 1 shows the removal of SS, COD, NH_3_–N, TN, and TP from the swine wastewater after the respective sections of the BDBO system. After the bio-coagulation process, it was found that SS, COD, NH_3_–N, TN, and TP could be removed effectively, with the removal efficiencies reaching 99.94%, 98.09%, 61.19%, 88.54%, and 99.92%, respectively. Guo and Ma [31] evaluated the potentials of bio-coagulation induced by *Rhodococcus erythropolis* for treating swine wastewater, in which the COD and NH_3_–N removal efficiencies were 45.2% and 41.8%, respectively. A study conducted by Ritigala et al. [32] also found that 89.1% of SS, 40.25% of COD, 7.82% of NH_3_–N, and 88.5% of TP in the swine wastewater could be removed by adding chemical flocculants (consisting of commercial PAC, magnetic seeds, and PAM). Compared to other studies, it was discovered that this bio–coagulation process performed better at removing SS, COD, and TP. Nevertheless, their NH_3_–N removal efficiency remained at the same level. With removal efficiencies of SS, COD, and TP of more than 85%, the bio–coagulation process induced by *Acidiphilium* sp. JZ6 and *A. thiooxidans* TS6 could be an effective method for enhancing the sedimentation ability of swine wastewater.

### 3.2. Dewatering Performance

The dewatering performance of the concentrated swine slurry after the bio–coagulation process was studied in the following investigations. As the dry solids (2.8–3.0% and 2.0–5.0%, respectively) and appearance (black and muddy state, respectively) of concentrated swine slurry and concentrated sludge are basically the same, the dewatering performance of the concentrated swine slurry can be evaluated by the SRF. Figure 2a shows the SRF of the concentrated swine slurry after the bio–coagulation process by adding the same number of bacterial suspensions, but different dosages of coagulant aids. Generally, the higher the SRF, the worse the sludge dewaterability is [33,34]. The SRF of the raw concentrated swine slurry without any conditioning was 17.0 × 10^12^ m/kg, which fell into the category of poor dewaterability. With the exception of the treatments of (i) with the single 0.2% bioflocculant and (ii) with the single 0.4% lime milk, the SRF values in the other treatments decreased, demonstrating that the dewaterability of the concentrated swine slurry can be significantly improved after the bio–coagulation process by adding different combinations of coagulant aids. Compared with treatments (iii) and (iv), it was found that the SRF slightly increased, from 0.35 × 10^12^ m/kg to 0.42 × 10^12^ m/kg, when the amount of bioflocculant increased to 0.3% and the amount of lime milk remained at 0.4%. Thus, 0.2% of the bioflocculant was selected for the subsequent treatments. When the dosage of lime milk was set to 0.3% in treatment (v), 0.4% in treatment (iii), and 0.5% in treatment (vi), the SRF value was 0.54 × 10^12^, 0.35 × 10^12^, and 0.32 × 10^12^ m/kg, respectively. Considering the SRF value and economic cost, the ratio of the bacterial suspensions and coagulant aids in treatment (iii), i.e., the combination of 5 mL of bacterial suspensions, 0.2% of the bioflocculant, and 0.4% of lime milk, was selected as the optimal alternative for the following pilot–scale dewatering experiment.

Figure 2b shows the pilot–scale dewatering performance of the concentrated swine slurry before and after the bio–coagulation process. The dewaterability performance improves with the amount of swine slurry treated. Only 3.8 L of raw concentrated swine slurry could be filtered through a pressure filter within 30 min. Moreover, after pressing and filtration, the swine slurry was paste–like and stuck to the press cloth rather than forming a “cake”. After the bio–coagulation dewatering process, a substantially enhanced filter pressing rate of concentrated swine slurry was achieved, of which 14 L of concentrated swine slurry could be completed within 20 min. This result validates the feasibility and effectiveness of the bio–coagulation process in improving the dewaterability of concentrated swine slurry. Additionally, after the bio–coagulation dewatering process, the raw concentrated swine slurry was shaped into the semi–dry cake, with the primary properties being characterized in Table 2. The moisture content and volume of the concentrated swine slurry decreased by around 40% and 90%, respectively, simultaneously achieving the goal of reducing the swine wastewater. The moisture contents and organic matter of the filter cake fluctuated in the ranges of 40–65% and 20–80%, respectively, satisfying the composting [35]. Notably, the filter cake, consisting of 49.43% of organic matter, 28.89 g/kg of TN, and 14.21 g/kg of TP, had potential as a fertilizer and would increase the revenue for pig farms through composting and farming. In addition, the concentrations of COD, NH_3_–N, TN, and TP in the filtrate after the bio–coagulation dewatering process further decreased (Table 1). The effluent SS, COD, and TP concentrations already met the discharge standard of GB18596–2001, significantly reducing the organic load of the subsequent biological treatment system and shortening the treatment process routes.

### 3.3. Bio–Oxidation Performance

One of the most popular reactors in the implementation of biological degradation with swine wastewater is SBR [36,37,38]. In this study, SBR was used to treat the filtrate after the bio–coagulation dewatering process, with the concentrations of COD and NH_3_–N being constantly monitored. It can be identified from Figure 3 that the influent COD and NH_3_–N concentrations were relatively low throughout the whole study period, ranging between 120–240 mg/L and 65–100 mg/L, respectively. The average concentrations of COD and NH_3_–N in the SBR effluent were 115 mg/L and 65 mg/L, respectively, which met the relevant discharge standard (GB18596–2001). Furthermore, the heavy metal ions commonly found in the swine wastewater, such as Cu, Zn, and As, were also monitored and met the national comprehensive discharge standard of sewage (GB8978–1996). Hence, compared to the original treatment process (HRT 15–20 d) in the pig farm [6], such SBR with HRT of 1 d could meet the requirements to reduce the process time and shorten the processing flow.

### 3.4. Implications of This Work

Economic feasibility is an essential issue in translating the BDBO system from the laboratory scale to a large–scale practical application. Both its simple configuration and low operation costs are crucial for the successful application of the BDBO system on a large scale. On one hand, the BDBO system would operate successfully with the help of the existing configuration, with the exception of the pressure filter. On the other hand, the operation costs of the BDBO system could be divided into two parts: inoculum/chemicals costs and energy consumption. According to our previous studies [6,22], the total costs of inoculum/chemicals were estimated to be 0.44 USD per cubic meter of wastewater (i.e., 0.44 USD/m^3^) as the inoculation size of 4.0% (*v*/*v*). In this study, with an inoculation size of 10% (*v*/*v*), the inoculum/chemicals costs were calculated to be 1.1 USD/m^3^. Moreover, the energy consumption was considered to be around 0.37 USD/m^3^ [6,22]. As a result, the overall operation cost of the BDBO system was estimated to be approximately 1.47 USD/m^3^. Compared with the original A/O + Fenton process adopted by this pig farm, the inoculum/chemicals costs and energy consumption were estimated to be 2.13 and 0.39 USD/m^3^, respectively. Worst of all, the NH_3_–N concentrations in the effluent of the original A/O + Fenton process were still between 130 and 305 mg/L [39]. Obviously, the overall operation cost was reduced by nearly 40%, confirming that the BDBO system is profitable and feasible. However, further study is needed to investigate the efficiency of the BDBO system in the face of stricter emissions standards, such as the European Union guidelines.

## 4. Conclusions

The feasibility of the BDBO system in treating swine wastewater with large amounts of refractory COD, TN, and TP attached to the SS was demonstrated for the first time. The bio–coagulation process showed high treatment efficiency, in which SS was completely removed, COD and TP were significantly reduced, and approximately 50% of NH_3_–N was removed. Accordingly, its dewaterability has been substantially improved with the dosages of bioflocculant and lime milk. In the subsequent bio–oxidation process, residual COD and NH_3_–N were further degraded and met the relevant discharge standard. Overall, this study provides a novel method with low operation costs and high efficiency for treating swine wastewater.

## Figures and Tables

**Figure 1 ijerph-20-02990-f001:**
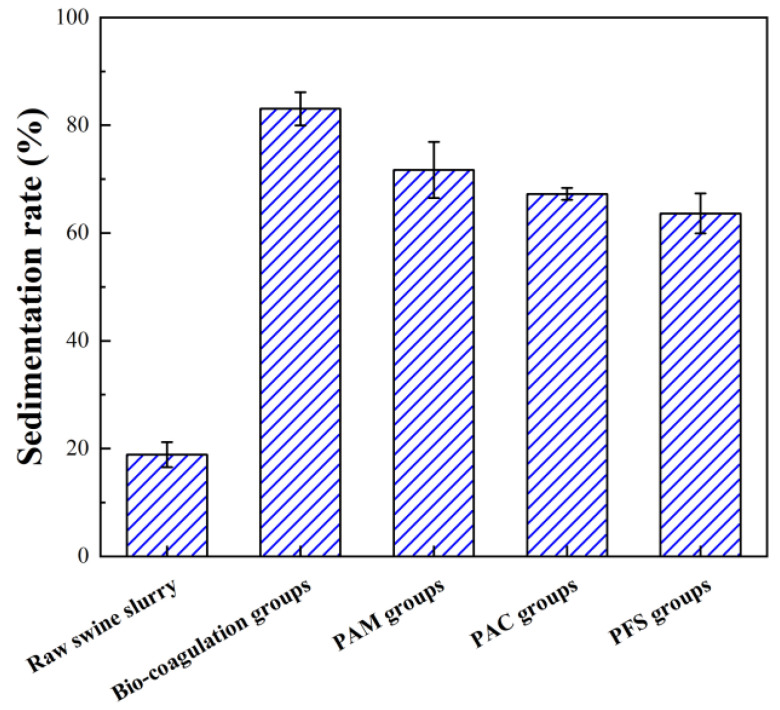
Sedimentation rate of swine wastewater after various coagulation treatments.

**Figure 2 ijerph-20-02990-f002:**
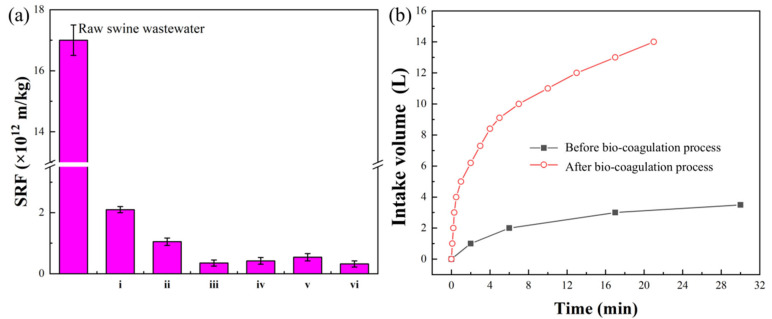
(**a**) SRF of the concentrated swine slurry after bio–coagulation with different dosages of dewatering agents and (**b**) intake volume of the swine slurry pumped into a filter press before and after the bio–coagulation process.

**Figure 3 ijerph-20-02990-f003:**
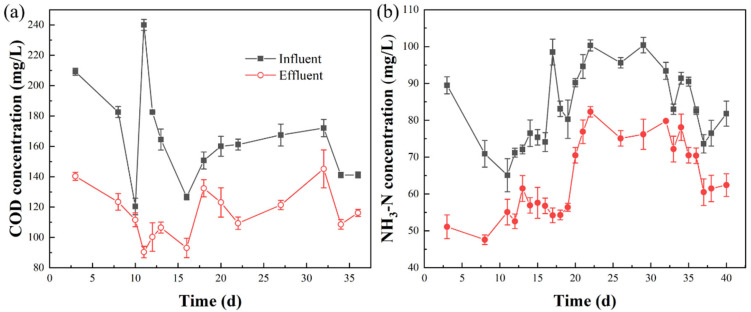
Profiles of COD (**a**) and NH_3_–N (**b**) during the long–term bio–oxidation process.

**Table 1 ijerph-20-02990-t001:** The primary properties of swine wastewater before and after various treatments (data provided by Hunan Environment Monitoring Centre, a professional third–party testing body).

Samples	SS (mg/L)	COD (mg/L)	NH_3_–N (mg/L)	TN (mg/L)	TP (mg/L)
Raw swine wastewater	13,520	7524	243	1065	1182
Effluent of bio–coagulation	8	144	94.3	122	0.94
Effluent of bio–oxidation	35	71	36.9	64.7	4.1
GB18596-2001 *	200	400	80	–	8

* GB18596-2001 refers to discharge standard of pollutants for livestock and poultry breeding, which was issued by Ministry of Ecology and Environment of the People’s Republic of China.

**Table 2 ijerph-20-02990-t002:** The primary properties of the semi–drying slurry cake after biological coagulation dewatering treatment.

	pH	Moisture Content	Organic Matter	TN (g/kg)	TP (g/kg)
Slurry cake	8.72	56.49%	49.43%	28.87	14.21

## Data Availability

Not applicable.

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
