# Peer review of "Feasibility of Bio–Coagulation Dewatering Followed by Bio–Oxidation Process for Treating Swine Wastewater"

_ijerph, 2023, doi:10.3390/ijerph20042990_

Round 1

Reviewer 1 Report

After reading this paper, the following concerns were drawn:

1.    Numerical results are needed in the abstract. Include values of quality parameters, “filtration resistance decreased from x to y”, English corrections are necessary (e.g., “After the treatment of BDBO”), etc.

2.    What do the authors mean by "inert COD"? Is it non-biodegradable organics? I think "refractory COD" or “recalcitrant organics" would be more appropriate.

3.    Item 2.1: what about BOD? Why did the authors not analyze BOD?

4.    Please, elaborate on the bio-coagulation process (introduction) and biocoagulant used in this study (material and methods).

5.    Why were these chemical coagulant dosages, and the inoculum ratio chosen?

6.    Authors are advised to include figures of experimental apparatus in the supporting document. 

7. Overall, material and methods section needs to be clarified. The heading should be renamed, and the overall organization of this section must be improved. Repeatability - can a person using the methods used by the authors repeat the study and obtain the same results? I do not.

8.    After reading the results section, it needs to be clarified that bio-coagulation (herein, named bio-flocculation) aimed to improve sedimentation capability, etc. Indeed, was sedimentation rate analysis based on what procedure? APHA, ASTM?

9.    Results and discussion needs more literature review for data comparison and analysis. In its current form, it needs more adequate assessment of the obtained data.

10. Findings from item 3.4 need to be more reliable. The authors presented data on economic costs without a proper methodological procedure for this estimation.

Reviewer 2 Report

At the end of the introduction, you could still better describe your aims to possibly get fertilizer or a raw matter for fertilizer or organic matter in the soil. That is presented at the end of the paper so your experiment is successful?  

The number of tests and the number of parallel analyses in all experiments?

Lines 103-104.  The Acidiphilum can use glucose as its carbon and energy sources, but Acidithiobacillus thiooxidans may use carbon dioxide as its carbon source but it will get its energy from the reaction: elemental sulphur  + oxygen -> sulphuric acid (thiooxidans means sulphur oxidation).  Instead of elemental sulfuret also other reduced sulphur compounds are possible. If the end product is sulphuric acid, the pH will be highly reduced.  In your reaction mixture lime milk and ammonium of swine slurry was alkaline. Anyhow what was the role of these bacteria used and what was the pH? Describe better in line 133 how you produced the bio-flocculants (in plural?).

Table 1: GB185-2001 should be explained. You explain this later in line 262. Is this standard a Chine one?  

Figure 2 b: before or after the process? Volume in which amount? The volume of solid parts or what?   The volume of the bio-coagulation group  (there is a printing error in the Figure text)  is much higher than that of raw slurry?  The figure should be independent of the text. Open better!

Describe better the flotation process!

Line 277 is the currency US $? 

Reviewer 3 Report

Please check the enclosed file, report the comments on a file, answer them and report the amended text

Round 2

Reviewer 1 Report

The weaknesses of previous report remain.  Research design is not appropriate, methods and results are not clearly presented, and conclusions are not supported by the results.

Reviewer 2 Report

Still in line 125: No organisms including Acidothiobacillus cannot use sulfur powder as its carbon source. Correct. 

Sulfur can serve as an energy source but not as carbon source. . 

Reviewer 3 Report

I still disagree with your choice of temperature
